# Reaching Syrian migrants through Dutch municipal registries for hepatitis B and C point-of-care testing

Chrissy P. B. Moonen [1,2]*, Elfi E. H. G. Brouwers[1,2], Christian J. P. A. Hoebe [1,2,3], Nicole H. T. M. Dukers-Muijrers [1,4], Jamila Bouchaara[1], Inge H. M. van Loo[3], Casper D. J. den Heijer [1,2]

1 Department of Sexual Health, Infectious Diseases and Environmental Health, Living Lab Public Health, South Limburg Public Health Service, Heerlen, The Netherlands, 2 Department of Social Medicine, Care and Public Health Research Institute (CAPHRI), Maastricht University, Maastricht, The Netherlands, 3 Department of Medical Microbiology, Infectious Diseases and Infection Prevention, Care and Public Health Research Institute (CAPHRI), Maastricht University Medical Center (MUMC+), Maastricht, The Netherlands, 4 Department of Health Promotion, Care and Public Health Research Institute (CAPHRI), Maastricht University, Maastricht, The Netherlands

* Chrissy.moonen@ggdzl.nl

**Data Availability Statement:** The data of this study contain potentially identifying and sensitive participant information. Due to the General Data Protection Regulation, it is not allowed to distribute

## Abstract

Undetected chronic hepatitis B virus (HBV) and hepatitis C virus (HCV) infections can lead to cirrhosis and liver cancer. Syrian migrants are the largest non-European migrant group in the Netherlands with HBV and HCV prevalence rates above 2%. This study aimed to reach Syrian migrants for HBV and HCV testing using point-of-care tests (POCT). A multifaceted strategy was employed to reach Syrian migrants aged ≥16 years from two Dutch municipalities for free-of-charge HBsAg and anti-HCV POCT using finger prick blood at the regional Public Health Service. All were personally invited by the Public Health Service by postal mail, based on municipal registry data. Respondents' medical history data were analysed descriptively and data on age, sex, and municipality were compared with non-participating invitees, using Pearson's Chi-square test. Of the study population (N = 832), 32.3% (n = 269) attended the testing. The mean age of participants was 36 years (range 16–70), 59.1% were men, and 66.5% were unemployed. Non-participation was higher in the younger age groups (<30 years) (p < .001). The POCT using finger prick blood was well received. None tested HBsAg or anti-HCV positive. With approximately one-third of participation, this study demonstrated relatively high reach of Syrian migrants for testing, compared to studies with similar recruitment methods. However, while the reach could be considered successful, testing failed to demonstrate new infection in this key population. Thereby, other methods may be preferred to identify new HBV and HCV infections, such as opportunistic testing within existing care processes.

## Introduction

Hepatitis is an inflammation of the liver that, if left untreated, can lead to liver cirrhosis and hepatocellular carcinoma (HCC) [1]. Hepatitis B virus (HBV) and hepatitis C virus (HCV)

or share any personal data that can be traced back (direct or indirect) to an individual. In addition, publicly sharing the data would not be in accordance with the participants' consent obtained for this study. Therefore, data used and/or analysed during the study are available from the head of the data archiving of the Public Health Service South Limburg on reasonable request. Interested researchers should contact the head of the data-archiving of the Public Health Service South Limburg (Tamara Kleine: tamara.kleine@ggdzl.nl) when they would like to re-use data.

**Funding:** This work was partly supported by the National Institute for Public Health and the Environment (https://www.rivm.nl/en). The funder had no role in study design, data collection and analysis, decision to publish, or preparation of the manuscript.

**Competing interests:** The authors have declared that no competing interests exist.

**Abbreviations:** HCC, Hepatocellular carcinoma; HBV, Hepatitis B virus; HCV, Hepatitis C virus; WHO, World Health Organization; DAA, Direct-acting antiviral agents; RIVM, The Dutch National Institute for Public Health and the Environment; PHS, Public Health Service; MEC-U, Medical Research Ethics Committees United; WMO, The Dutch Medical Research Involving Human Subjects Act; POCT, Point-of-care testing; GP, General practitioner; HBsAg, Hepatitis B surface antigen; EU, European Union; Anti-HCV, Hepatitis C virus antibodies; FDA, Food and Drug Administration; CE, Conformité Européenne; MUMC+, Maastricht University Medical Centre; EDTA, Ethylenediaminetetraacetic acid; SPSS, The Statistical Package for the Social Sciences; HIV, Human immunodeficiency virus.

contributed to almost 98 per cent of all hepatitis-related deaths in 2017 [2]. An estimated 1.1 million deaths were attributable to these viruses in 2019 worldwide [3]. The World Health Organization (WHO) has called for action to eliminate hepatitis by the year 2030 [4]. Micro-elimination by testing high-risk groups is key to addressing hepatitis as a public health problem, especially given the often asymptomatic course of HBV and HCV [5]. Once the infection is identified, care can be initiated with treatment options that have greatly improved in recent years, particularly with the development of direct-acting antiviral (DAA) agents for HCV [6].

In response to the WHO call, the Dutch National Institute for Public Health and the Environment (RIVM) developed a National Hepatitis Plan in 2016 [7]. Major pillars in this plan are early infection detection and treatment, improved organisation of hepatitis care, and national registration [7]. Despite the low incidence of acute HBV, chronic HBV, and acute HCV (0.6, 6, and 0.36 per 100.000 respectively) in the Netherlands [8], testing high-risk groups can further reduce the burden of hepatitis. In the Netherlands, the 25 regional Public Health Services (PHS) are responsible for public health, prevention, and health promotion at the municipal level.

Since first-generation migrants account for most Dutch chronic HBV (81%) and HCV (60%) cases, targeted testing initiatives for this group should be explored [9]. In the Netherlands, Syrian migrants are the largest non-European risk group with HBV and HCV prevalence levels above 2%, indicating that Syrian migrants are a relevant target group for testing [10, 11]. Combined HBV and HCV testing of migrants in a low-endemic country such as the Netherlands has been shown to be cost-effective [12]. However, high-risk groups such as migrants are often not well reached for testing and treatment [13].

This study is part of a nationally coordinated implementation research to reach first-generation non-European migrants for HBV and HCV testing, following the National Hepatitis Plan. In the present study, we aimed to 1) reach Syrian migrants in two Dutch municipalities by inviting them through municipal registries and 2) identify HBV and HCV infections in Syrian first-generation migrants by point-of-care testing (POCT) using finger prick blood and to rapidly link infected individuals to care.

# Materials and methods

## Study design and setting

In this cross-sectional study, Syrian migrants from two Dutch municipalities (Heerlen and Maastricht) were invited by the PHS of South Limburg for free HBV and HCV POCT using finger prick blood at a local PHS site. The municipalities of Heerlen and Maastricht are home to the highest number of Syrian migrants in the region of South Limburg. The Medical Research Ethics Committees United (MEC-U) agreed that the research was outside the scope of the Dutch Medical Research Involving Human Subjects Act (WMO) (reference number W20.054).

## Study population

The study population consisted of Syrian status holders living in the municipalities of Heerlen and Maastricht in the Netherlands, identified through municipal registries. Only Syrian migrants (born in Syria) aged 16 years or older were invited for testing.

## Multifaceted strategy

The strategy employed in this study was established by combining various elements, inspired by a comprehensive literature review and the insights gained through conducted focus groups

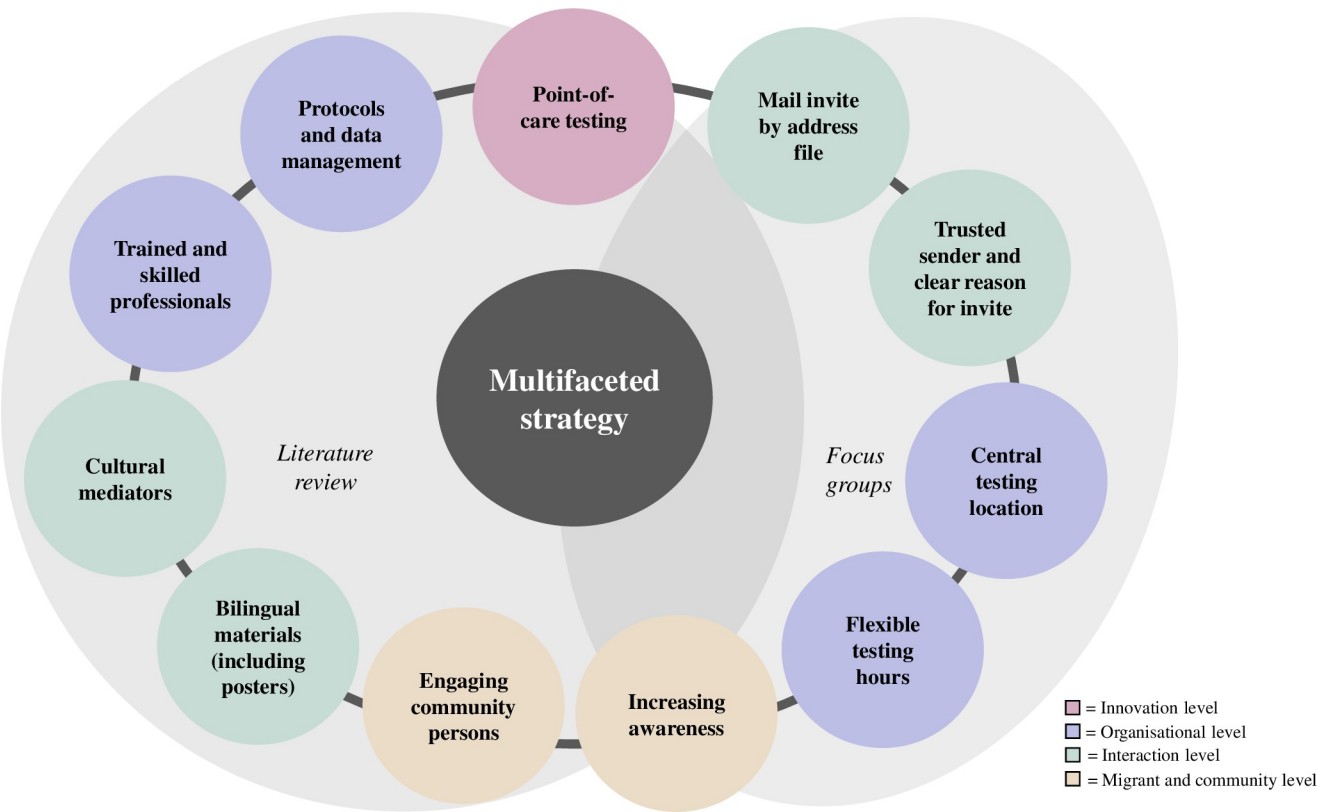

**Fig 1. Components of our multifaceted strategy based on literature and focus groups.**

before commencing the study (Fig 1) [14]. The elements were structured into several levels, namely the innovation level, the organisational level, the interaction level, and the migrant and community level. These elements were combined with the aim of eliminating barriers and facilitating participation in the testing.

A Syrian physician employed by the PHS undertook focus groups with Syrian migrants, conducting separate sessions for men and women. The outcomes of these sessions revealed the participants' preference for an invite by mail through an address file, sent by a familiar governmental organisation that provided a clear rationale for the free testing. The invitation letter explained that asymptomatic infections may go undetected and that testing could clarify the infection status and enable timely treatment. It also stressed the importance of protecting others from the infectious nature of HBV and HCV and noted the higher prevalence of these viruses in the country of birth as the reason for the invitation. The municipality was perceived as reliable, attributed to its role in providing housing for migrants. Therefore, the migrants were invited by letter with a clear reason for the invitation from the PHS—an organisation clearly related to the municipality. Furthermore, the participants of the focus groups preferred testing at a conveniently located place and possibilities for evening testing. We complied with this by organising testing at easily accessible PHS sites, reachable by public transportation and ensured testing opportunities during both afternoon and evening hours.

Besides facilitating factors, several barriers were pointed out in the focus groups, including the language gap and lack of knowledge about the viruses. At the interaction level, the language barrier was tackled by providing bilingual materials (in Dutch and Arabic) in B1 reading level. Additionally, translators and cultural mediators were present throughout the testing days,

offering linguistic and transcultural mediation, clarifying information, and answering questions. These roles were undertaken by native and culturally sensitive peers and care professionals. To address knowledge gaps, information on the viruses and the testing opportunity was incorporated into the invitation letters, featured on the PHS website, and displayed on bilingual posters disseminated across locations regularly frequented by Syrians, including supermarkets, local mosques, community centres, grillrooms, and general practitioners' offices.

The systematic literature review conducted before this study endorsed addressing the language barrier, raising awareness, and ensuring an accessible testing process. In an effort to enhance visibility and knowledge, local mosques were engaged to encourage community participation in testing. Moreover, the review highlighted POCT as an innovation to potentially overcome testing barriers, facilitating prompt identification of infections and timely linkage to care for treatment initiation [15]. Certified, registered, and trained nurses administered the tests with strict adherence to protocols to maintain a systematic and streamlined approach. Data management was conducted using standardised forms to ensure consistency and accuracy throughout the study. To limit the use of personal data and safeguard participant privacy, each invitee was assigned a unique participant code, which was used to link their tests to their respective identities. During registration, PHS staff explained the use of a participant code to the participants.

## Study protocol

**Pre-testing.**   Name and address information of potential participants were requested from the municipal registries to send invitation letters. Stakeholders, including general practitioners (GPs) in the municipality and the gastrointestinal-liver department of the regional hospitals, were informed of the study. Subsequently, the study population received an invitation for voluntary, free, POCT for HBV and HCV.

**Testing.**   The testing days took place on the 13th and the 15th of March 2023 in Heerlen and on the 8th and 10th of May 2023 in Maastricht. These testing dates were selected to avoid the Islamic fasting month of Ramadan. Non-respondents received an invitation for an alternative testing day on the 24th of April for Heerlen and the 24th of May for Maastricht.

Before testing, participants completed a written informed consent form with a unique participant code and a medical history questionnaire containing sociodemographic, behavioural and medical questions. After this, the nurse performed the finger prick for the POCT. Participants could wait for their test results in a waiting area, where drinks, treats and toys for children were provided. After twenty minutes, the test results were assessed and communicated by a PHS physician to the participant. Regardless of the test results, the participants' data were recorded in the medical records in accordance with legal requirements. In the case of a positive test result, a direct referral to care was embedded, including confirmation of the test result (serology via venepuncture), and if confirmed positive notification of the GP, referral to the hospital, and source and contact investigation.

## Data collection and analysis

**Pre-testing reason(s) for non-participation.**   Invitees who were unwilling or unable to participate could anonymously indicate their main reason(s) for non-participation through a QR code or link provided in the invite and reminder. This led to a dual-lingual (Dutch/Arabic) one-question online questionnaire into market research software Crowdtech (ISO-20252 and ISO-27001 certified, London, UK). Twenty-three fixed answer options, which were inspired by the systematic literature review conducted before this study, and one open response option were given [14]. After testing, data were extracted and analysed descriptively.

**Questionnaire.** Before testing, participants completed a medical history questionnaire on demographics, the main reason for testing, hepatitis (vaccination) status and risk assessment. Participants could again choose a Dutch or an Arabic version. After the testing, the paper questionnaires were entered (translated for the Arabic version) into Crowdtech by the lead researcher (CM). After completion of the pilot study, data were extracted and ten percent of the data were randomly checked for errors by a management assistant. The education level was categorised according to the education system in Syria as not educated, practically educated (basic education certificate), combined educated (technical/vocational secondary education certificate, technical diploma certificate, certificate of associate degree/licensed assistant, and general secondary education certificate), or theoretically educated (academic bachelor or master). Employment status was categorised as working (part-time or full-time), non-working (unemployed, benefits, housewife/man, and retired) and student. Marital status was reduced to partner (married, unmarried partner) or no partner (no partner, widowed, divorced).

**Point-of-care testing.** A finger prick for a capillary blood sample was performed by a nurse for HBV and HCV POCT. The presence of hepatitis B surface antigen (HBsAg) was tested according to the manufacturer's protocol using DETERMINE™ HBsAg 2 by Abbott (Chicago, United States), which meets the requirements of the European Union (EU) and WHO for analytical sensitivity. We chose this test for its good clinical performance in validation studies [16, 17]. The WHO evaluated this test with a specificity and sensitivity of 100 (95% CI 98.8–100) and 100 (95% CI 98.2–100) respectively [18]. The test demonstrated user-friendliness with rapid results, entailed lower costs, and required fewer trained professionals, infrastructure, and logistical resources compared to laboratory testing [16, 17, 19]. HCV antibodies (Anti-HCV) were tested according to the manufacturer's protocol with the Food and Drug Administration (FDA) approved OraQuick® HCV Rapid Antibody Test by OraSure (Bethlehem, United States) [20]. As well as the Determine HBsAg 2, the OraQuick Anti-HCV test demonstrated good sensitivity and specificity in validation studies [20–24]. The WHO validation confirmed the test's specificity at 99.7% (95% CI 98.3–100.0%) and sensitivity at 100% (95% CI 97.8%–100.0%) [25]. Both tests hold the CE certification, signifying their compliance with safety standards for sale and use in the European Economic Area, as evaluated by the Conformité Européenne (CE). Positive POCT results were confirmed by the Department of Medical Microbiology, Infection Prevention and Infectious Diseases of the Maastricht University Medical Centre (MUMC+) using HBsAg II and anti-HCV screening (Elecsys, Roche, Switzerland) on venous blood collected in Ethylenediaminetetraacetic acid (EDTA) blood tubes.

**Comparison of participants with non-participants.** Participants were compared with invited non-participants based on sex, age group, and municipality of administration of the test using the Pearson Chi-Square test ($X^2$) and multivariable analyses. The Statistical Package for the Social Sciences (SPSS, version 27.0, IBM, Armonk, USA) for Windows was used. A p-value of 0.05 or less was considered as statistically significant.

## Results

In total, 269 individuals were tested in Heerlen (n = 156) and Maastricht (n = 113) (Fig 2). Approximately five percent of the invitations were undeliverable. No HBsAg or anti-HCV was detected.

## Study population and characteristics

The study population consisted of those who received the invitation and those eligible for testing, although they were not initially invited (N = 832). Some individuals (n = 23) were accidentally not invited, probably due to outdated information from the municipal registries, but accompanied relatives who were invited and were tested anyway. Ten of the respondents who

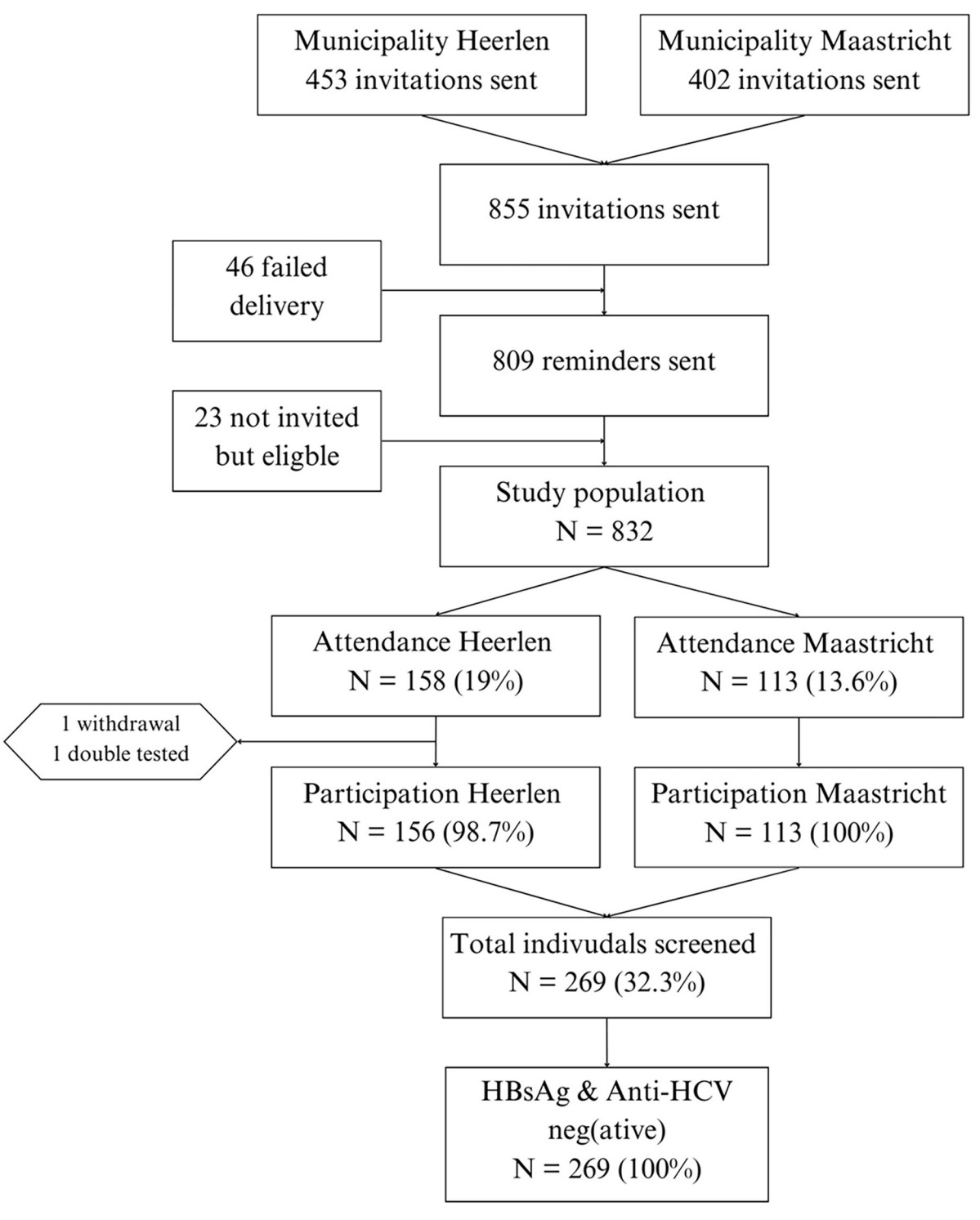

**Fig 2. Flowchart of participation and test results.**

were tested turned out not to be within the definition of the study population. Their test results were excluded from the analysis.

The majority of participants lacked Dutch language proficiency and therefore completed the Arabic questionnaire (72.1%) and needed translation during testing and communication of the test results. Conversations with participants revealed that POCT using finger prick blood was well received, mostly because of the minimally invasive finger prick and short turnaround time of approximately 20 minutes for the test results. The nurses expressed their satisfaction with the ease of administering the POCT and appreciated the support provided by the cultural mediators.

The mean age of the participants was 35.7 years (range: 16–70 years) (Table 1). Most participants were men (59.1%, n = 159) and non-working (66.5%, n = 177). The majority of participants had a partner (58.4%, n = 150) and 47.5% (n = 124) declared that they lived with a partner. On average, participants had lived in Syria for 26.9 years, and most mothers (99.6%, n = 268) and fathers (97.0%, n = 259) of the participants were born in Syria as well.

### Previous HBV-vaccination and hepatitis status

Of the participants, 46.3% were unaware of their HBV vaccination status. A small proportion (15.3%) was certain about being vaccinated previously, and 38.4% were not vaccinated. Twelve participants (4.7%) indicated that they have (had) hepatitis; however, the majority of them (83.3%) did not know what type. However, past HCV infections were not considered, as these would have been detected through the anti-HCV testing. Of the (previous) infected individuals, three (25%) were treated.

### Risk factors

Regarding risk factors for HBV and HCV, 25.3% (65/257) reported having undergone surgery abroad, 3.1% (8/257) indicated having donated an organ, and 5.1% (13/255) said they had used drugs intravenously. It is worth noting that the relatively high percentage claiming that they had used drugs intravenously may be explained by some participants' interpretation that drugs meant the same thing as medication. Therefore, the Arabic translation was revised following the second session. None (0/259) reported positivity for human immunodeficiency virus (HIV) or a history of sexual contact with a person of the same sex.

### Reasons to (not) get tested

The most common reason for participation was being invited (n = 194). Other often-recurring reasons were 'I want certainty' (n = 64) and 'I want to be sure I do not have it' (n = 58).

Those who did not want to attend could indicate their main motive(s) anonymously online before testing. Twenty-three individuals declared their motives for not participating, and most recurring reasons were not feeling sick/not having any symptoms (n = 9), not having time (n = 9), and already tested (n = 4). Other indicated reasons were a lack of knowledge about hepatitis (n = 3), lack of transportation (n = 2), lack of risk perception (n = 1), experienced language barrier (n = 1), lack of faith in the PHS (n = 1), being vaccinated (n = 1), and afraid to get infected on site (n = 1). Again, the vast majority of the respondents completed the non-response question in Arabic (91.3%).

### Comparison of participants and non-participants

Participants and non-participants did not differ with regard to sex and municipality of residence, but did differ regarding age (Table 2). Non-participation was higher in the 16–19 and 20–29 age groups (Fig 3).

**Table 1. Descriptive demographic characteristics.**

| Characteristics, N = 269[a] | n (%) |
|---|---|
| **Age in years** | |
| Mean (range), s | 35.7 (16–70), 13.3 |
| 18–19 | 37 (13.8) |
| 20–29 | 62 (23.0) |
| 30–39 | 67 (24.9) |
| 40–49 | 54 (20.1) |
| 50–59 | 38 (14.1) |
| 60+ | 11 (4.1) |
| **Sex[b]** | |
| Male | 159 (59.1) |
| Female | 110 (40.9) |
| **Level of education[c]** | |
| Uneducated | 28 (10.5) |
| Practically educated | 63 (23.6) |
| Combined educated | 126 (47.2) |
| Theoretically educated | 50 (18.7) |
| **Employment[d]–multiple answers possible** | |
| Working | 75 (28.2) |
| Non-working | 177 (66.5) |
| Student | 60 (22.6) |
| **Relation status[e]** | |
| Partner | 150 (58.4) |
| No partner | 107 (41.6) |
| **Living situation–multiple answers possible** | |
| Alone | 55 (21.1) |
| With partner | 124 (47.5) |
| With children | 81 (31.0) |
| With parents | 63 (24.1) |
| **Years lived in Syria** | |
| Mean (range), s | 26.9 (1–63), 14.4 |
| **Country of birth mother** | |
| Syria | 268 (99.6) |
| Other | 1 (0.4) |
| **Country of birth father** | |
| Syria | 259 (97.0) |
| Other | 8 (3.0) |

[a] Missing data in variables: level of education n = 2 (0.7%), employment n = 3 (1.1%), relation status n = 12 (4.5%), living situation n = 8 (3%), years lived in Syria n = 13 (4.8%), and country of birth father n = 2 (0.7%).

[b] Self-reported, no declared non-binary.

[c] Practically educated: basic education certificate. Combined educated: technical/vocational secondary education certificate, technical diploma certificate, certificate of associate degree/licensed assistant, or general secondary education certificate. Theoretically educated: academic bachelor or master.

[d] Working: working part-time or full-time. Non-working: unemployed, benefits, housewife/man, retired.

[e] Partner: married or unmarried partner. No partner: no partner, widowed, or divorced.

**Table 2. Characteristics of the participants and non-participants.**

|  | Participants n (% within participation) | Non-participants n (% within participation) | P-value |
|---|---|---|---|
| **Sex** |  |  | .516 |
| Male | 159 (31.5) | 346 (68.5) |  |
| Female | 110 (33.6) | 217 (66.4) |  |
| **Municipality** |  |  | .064 |
| Heerlen | 156 (35.1) | 288 (64.9) |  |
| Maastricht | 113 (29.1) | 275 (70.9) |  |
| **Age group** |  |  | < .001 |
| 16–29 years | 100 (23.4) | 328 (76.6) |  |
| 30–70 years | 169 (41.8) | 235 (58.2) |  |

## Discussion

In this study, we aimed to reach Syrian migrants for HBV and HCV POCT using finger prick blood by a multifaceted strategy, including a postal invite. For this method, we achieved a relatively high participation rate (32.3%), compared to studies with similar recruitment methods. For example, two Spanish studies in which the general Spanish population was invited by personal letter for HCV testing, showed participation rates of 11.2% and 4.1%, respectively [26, 27]. Compared to these Spanish studies, the participation rate was slightly higher in another Dutch project (17.8%), in which 10,000 migrants were invited per letter by the PHS [28]. The higher participation rate in the current study may be attributed to the multifaceted strategy, which was developed based on the outcomes of focus groups and an extensive literature review of facilitators and barriers for testing [14]. This hypothesis is supported by the 29.7%

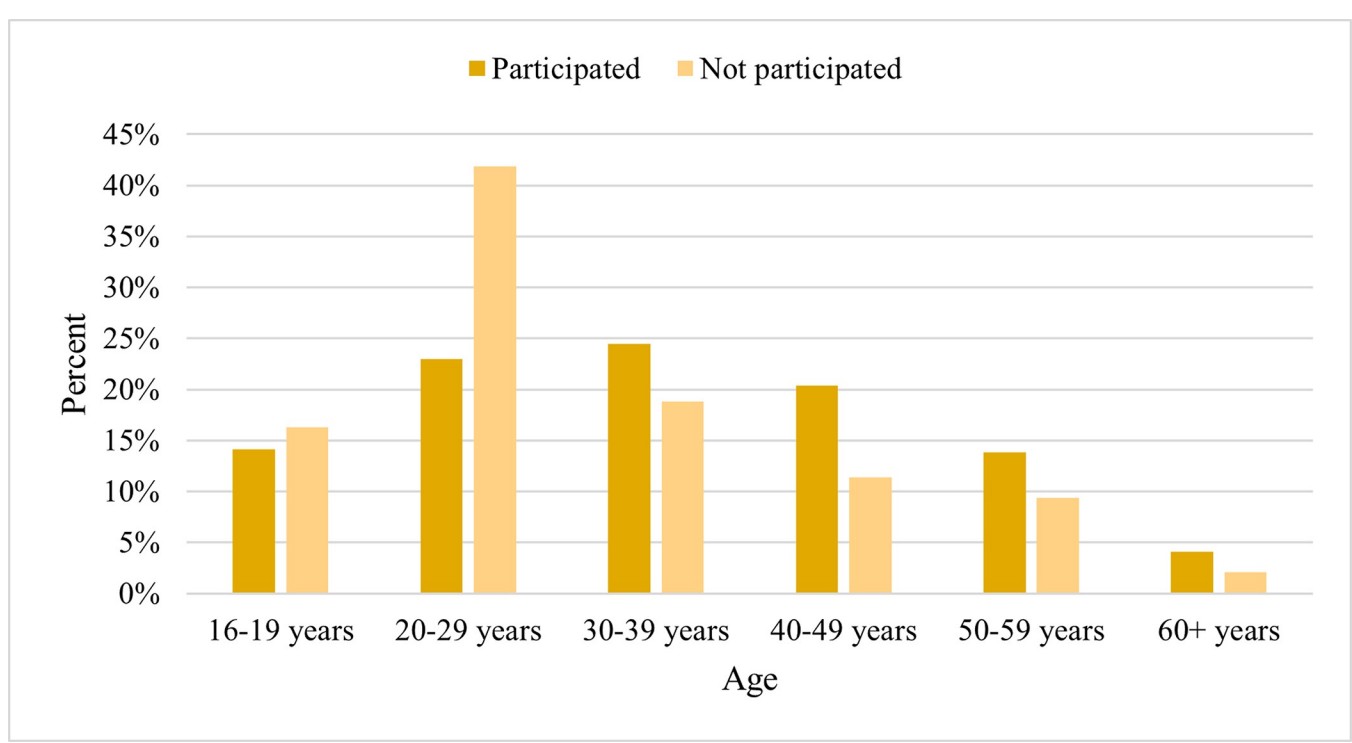

**Fig 3. Bar chart of (non)participation per age group (N = 832).**

participation rate in another Dutch study [29], in which also additional efforts were made to recruit migrants from Afghanistan, Iran, Iraq, the former Soviet Republics, and Vietnam, in addition to a personal invitation.

The usefulness of this strategy is still considered limited since HBV and HCV infections were not detected. HBV and HCV prevalence for both infections is suggested to be over 2% in Syrian migrants [10, 11] and the participants did report additional risk factors. However, 4.7% of participants reported having (had) hepatitis, indicating that the diagnosis had already been made previously, for example, in Syria during the mandatory premarital screening, or elsewhere in the Netherlands. Other possible explanations for the absence of present infections in the current study include clearance of past (HBV) infections or potential infections, whether detected or undetected, among non-participants. Nevertheless, we lack certainty due to non-participation and lack of information about cleared HBV infections, as the HBV POCT did not provide information on this.

The participation in testing was higher among invitees over 30 years old than among younger invitees. Although age (distribution) is often underreported in prevalence studies, older age and male sex are associated with higher HBV prevalence rates [30, 31]. This is supported by a recent study in which Syrian blood donors aged 18–25 years were less likely to be infected [32]. Besides, a Dutch study showed that younger people were more likely to be immunised by vaccination and less likely to be immunised by infection [33].

In this study, POCT using finger prick blood was well received by both participants and staff, aligning with previous findings [34]. Nursing staff found the tests easy to administer, which was echoed by the successful execution of all tests. Compared to natives, migrants may face more challenges in accessing healthcare, for example, due to linguistic barriers or discrimination [35]. This indicates that the optimal use of a care moment is even more valuable. POCT has the potential to improve the cascade of care and reduce loss to follow-up [35]. Moreover, the major demand for the Arabic questionnaire and the indispensability of a cultural mediator emphasize the importance of bilingual (informational) materials and (transcultural) translators during testing. We anticipate that the insights from this study hold considerable generalisability and can serve as an indication for other countries or regions adopting a similar approach for this population.

## Limitations

Although name and address details were requested shortly before the invitations were sent out, the five percent undeliverable letters suggest that municipal records data are not sufficiently updated. Events that may have influenced testing participation included a regional transportation strike and elections for provincial councils and water boards (March 15, 2023) and a violent conflict in Maastricht between two groups of Syrian youths that made headlines (April 29, 2023). As noted by the Syrian (transcultural) translator involved in this pilot, the earthquake in Syria and Turkey in February 2023 caused significant emotional distress due to the loss of family and friends in the homeland. This and other concerns due to the earthquake may have also affected participation.

As for the pilot itself, implementation was labour-intensive, requiring careful planning of staff. Furthermore, testing was offered only to the Syrian migrant group among several migrant groups in two of the sixteen municipalities in the South-Limburg region. The questionnaire was self-administered, making it prone to bias, for example, because of the fear of stigmatization. Due to the absence of identified infections, we cannot draw conclusions regarding the risk factors for infection or evaluate follow-up care as we initially intended. Furthermore, the strategies have not been evaluated in an experimental setting, so the individual

or combined effects are unknown. However, the design and lessons learned from this study can be valuable in setting up other migrant HBV and HCV testing projects.

## Recommendations

For testing to best fit the target audience, we recommend exploring this with the target audience beforehand through, for example, focus groups. Considering the absence of infections in this study and the time and money intensity of targeted testing, we recommend a more sustainable approach by implementing HBV and HCV testing in existing care processes (opportunistic testing). However, the people reached during targeted testing may not be those who are seen in regular care. Therefore, targeted testing might still be a useful complementary action for reaching populations in the highest prevalence segment, for example >10% for HBV and >3% for HCV when looking at the global distribution maps of CDA Foundation's Polaris Observatory [11]. As in most immigrant populations, language barriers were present, and hence, translated materials and translator/cultural mediators are essential when communicating with migrants. We also recommend greater diversity in the origins of staff in healthcare organisations to better serve individuals from other countries of birth. Furthermore, POCT can overcome miscommunication regarding the need for follow-up and reduce frequent loss to follow-up in target populations as multiple visits for diagnosis are unnecessary. Testing for multiple infectious diseases simultaneously by POCT can improve the cascade of care and may expedite the linkage to care [36].

## Conclusions

With about one-third participation rate, we can conclude that a multifaceted strategy for HBV and HCV testing, including an invite by letter, is a good way to reach migrants for testing, but additional efforts are needed to reach more people at risk. Although no infections were detected, POCT using finger prick blood was well received and could yield benefits for hepatitis programs and patients.

## Acknowledgments

Special thanks to the dedicated staff members who provided assistance before and during the testing period, and we extend extra gratitude to the translators for their work in providing translations. Furthermore, we would like to acknowledge the Medical Microbiology lab of Maastricht University Medical Centre+ for their expertise and their medical microbiological advice. We extend our appreciation to the PHS of Groningen and Amsterdam for their willingness to share their thoughts and experiences during the implementation study. The authors acknowledge the research infrastructure provided by the Dutch Collaborative Academic practice for public health Infectious diseases (CAPI). Lastly, the main author wants to thank the co-authors for their efforts and support.

## Author Contributions

**Conceptualization:** Chrissy P. B. Moonen, Elfi E. H. G. Brouwers, Christian J. P. A. Hoebe, Nicole H. T. M. Dukers-Muijrers, Casper D. J. den Heijer.

**Formal analysis:** Chrissy P. B. Moonen.

**Funding acquisition:** Christian J. P. A. Hoebe, Casper D. J. den Heijer.

**Project administration:** Chrissy P. B. Moonen, Elfi E. H. G. Brouwers, Christian J. P. A. Hoebe, Nicole H. T. M. Dukers-Muijrers, Jamila Bouchaara, Casper D. J. den Heijer.

**Supervision:** Christian J. P. A. Hoebe, Nicole H. T. M. Dukers-Muijrers, Casper D. J. den Heijer.

**Writing – original draft:** Chrissy P. B. Moonen.

**Writing – review & editing:** Elfi E. H. G. Brouwers, Christian J. P. A. Hoebe, Nicole H. T. M. Dukers-Muijrers, Jamila Bouchaara, Inge H. M. van Loo, Casper D. J. den Heijer.

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
