## [Decision Letter · Decision Letter 0]

16 Jul 2024

PONE-D-23-37923Reaching Syrian migrants through Dutch municipal registries for hepatitis B and C point-of-care testingPLOS ONE

Dear Dr. Moonen,

Thank you for submitting your manuscript to PLOS ONE. After careful consideration, we feel that it has merit but does not fully meet PLOS ONE’s publication criteria as it currently stands. Therefore, we invite you to submit a revised version of the manuscript that addresses the points raised during the review process.

We look forward to receiving your revised manuscript.

Kind regards,

Livia Melo Villar

Academic Editor

PLOS ONE

2. In the online submission form, you indicated that [The data of this study contain potentially identifying and sensitive participant information. Due to the General Data Protection Regulation, it is not allowed to distribute or share any personal data that can be traced back (direct or indirect) to an individual. In addition, publicly sharing the data would not be in accordance with the participants’ consent obtained for this study. Therefore, data used and/or analysed during the study are available from the head of the data archiving of the Public Health Service South Limburg on reasonable request. Interested researchers should contact the head of the data-archiving of the Public Health Service South Limburg (Tamara Kleine: tamara.kleine@ggdzl.nl) when they would like to re-use data.]. 

Additional Editor Comments:

Dear Author,

I have read the paper and comments of the reviewers. After careful reading, I observed that sampling strategy adopted by the authors is not so good, the optimal alternative should be RDS, Starfish, etc. However, authors should explain why they choose this sampling method, they may show that a less than optimal technical choice could be better once the relative costs are taking in consideration. Please also read and answered the reviewer's comments.

Sincerely,

Livia

Reviewers' comments:

Reviewer's Responses to Questions

**Comments to the Author**

1. Is the manuscript technically sound, and do the data support the conclusions?

Reviewer #1: Yes

Reviewer #2: Yes

2. Has the statistical analysis been performed appropriately and rigorously? 

Reviewer #1: N/A

Reviewer #2: Yes

3. Have the authors made all data underlying the findings in their manuscript fully available?

Reviewer #1: Yes

Reviewer #2: Yes

4. Is the manuscript presented in an intelligible fashion and written in standard English?

Reviewer #1: Yes

Reviewer #2: Yes

5. Review Comments to the Author

Reviewer #1: This well written manuscript describes a study among Syrian migrants in two Dutch cities. The aim was to develop an outreach strategy to screen them by measuring antibodies in the blood, against hepatitis B virus (HBV) and hepatitis C virus (HCV). The Syrian migrants were chosen because it was suspected that these infections were more highly prevalent than among other Dutch residents.

I have some minor comments to further improve the manuscript.

Minor comments

In the introduction two aims are stated (lines 74-75): 1. how to best reach the Syrian migrants in two Dutch municipalities and 2. To identify HBV and HCV infections in these migrants.

For the second aim I suggest to add everywhere in the manuscript that finger prick blood is used next to using a POCT since these two are not coupled and both are deviant from routine testing settings.

Abstract

Line 26: please add ‘(POCT)’ after ‘testing’, since this abbreviation is used later in line 32.

It is also good to add ‘using finger prick blood’ after ‘POCT’.

Multifaceted strategy

Line 127: since there are very many POCT on the market, I suggest to replace ‘the’ by ‘a’ before ‘point-of-care testing’.

Results

Legend of Fig 2. (line 220): I suggest to add ‘and test results’

Line 234: Please provide some more information on ‘short turnaround time’: what was the median time and range in minutes?

Discussion

Line 300: please adjust and do not start a new sentence. So ‘was made, for example….’.

Line 303: please adjust to: ‘non-participants’.

Line 326: Please replace ‘whom’ by ‘who’.

Line 332: Please adjust to ‘successful’.

Limitations

Although the DETERMINE test for HBsAg and the OraQuick assay for HCV antibody detection are both stated to have a very high sensitivity, would you still consider using these in future settings on finger prick blood?

Also, I think it is a pity that no tests seem to have been performed concomittantly on regular EDTA blood samples (as was described to be performed in case confirmation was needed) and in the finger prick blood samples with the POCTs, to have an impression of the actual validity of the high sensitivity of these POCTs in this setting.

Reviewer #2: Review Report

Journal: PLOS ONE

Manuscript Number: PONE-D-23-37923

Article Type: Research Article

Title: Reaching Syrian migrants through Dutch municipal registries for hepatitis B and C point-of-care testing

Authors: Chrissy PB Moonen, Elfi EGH Brouwers, Christian JPA Hoebe, Nicole HTM Dukers-Muijrers, Jamila Bouchaara, Inge HM van Loo, Casper DJ den Heijer.

In this manuscript, the authors aimed to reach Syrian migrants as part of a nationally coordinated implementation research to reach first-generation non-European migrants for HBV and HCV screening using point-of-care testing in two Dutch municipalities (Heerlen and Maastricht). The article described a solid and very well-presented research which is absolutely consonant with OMS agenda for viral hepatitis elimination by 2030.

Overall, the study had great merit in the detailed presentation of the multifaceted strategies used that made it possible to reach a significant number of migrants (approximately 1/3 of the individuals invited to participate), an extremely valuable information that could be implemented in similar studies in different parts of the world in an effort to achieve the microelimination of viral hepatitis in key populations. Although previous studies had reported HBV and HCV infection rates over 2% in Syrian migrants, no positive cases were detected within the 269 individuals screened. Nonetheless, based on a previous systematic literature review that supported the multifaceted strategies proposed by the authors, the research efforts fulfilled their objective of screening a difficult-to-reach population, addressing one of the main bottlenecks for the elimination of viral hepatitis.

The quality of the written English is excellent thorough out the article and, in my opinion, only a few minor revisions are necessary for publication:

1. Page 13, line 277: A parenthesis ")" mistakenly appeared before the beginning of the sentence.

2. Page 13, line 278: An "n" was missing before “=3” in the parenthesis showing the number of people who reported lack of knowledge about hepatitis.

3. Page 13, line 291: There is an error in the caption of figure 3: part(non)participation is written instead of (non)participation.

4. Page 15, lines 331 and 333: I believe the in-text citations of references 34 and 35 are not correctly presented and should be corrected.

6. PLOS authors have the option to publish the peer review history of their article (what does this mean?). If published, this will include your full peer review and any attached files.

Reviewer #1: **Yes: **Sylvia M. Bruisten

Reviewer #2: No

---

## [Author Response · Author response to Decision Letter 0]

30 Jul 2024

Dear Editorial Office and reviewers, 

We thank you for considering our manuscript and for your positive and valuable comments. The revised manuscript, with track changes, has been resubmitted. This document contains the response to comments.

Reviewer 1

This well written manuscript describes a study among Syrian migrants in two Dutch cities. The aim was to develop an outreach strategy to screen them by measuring antibodies in the blood, against hepatitis B virus (HBV) and hepatitis C virus (HCV). The Syrian migrants were chosen because it was suspected that these infections were more highly prevalent than among other Dutch residents.

I have some minor comments to further improve the manuscript.

Comment 1. In the introduction two aims are stated (lines 74-75): 1. how to best reach the Syrian migrants in two Dutch municipalities and 2. To identify HBV and HCV infections in these migrants. For the second aim I suggest to add everywhere in the manuscript that finger prick blood is used next to using a POCT since these two are not coupled and both are deviant from routine testing settings. 

Reply: Thank you for your positive feedback and valuable comments. ‘’using finger prick blood’’ has been added throughout the manuscript after POCT (line 33, 77, 84, 236, 299-300, 335, 392).

Abstract 

Comment 2. Line 26: please add ‘(POCT)’ after ‘testing’, since this abbreviation is used later in line 32.

Reply: Thank you for the thoughtfulness. (POCT) has been added after testing (line 26).

Comment 3. It is also good to add ‘using finger prick blood’ after ‘POCT’.

Reply: We agree with this suggestion. ‘using finger prick blood’ has been added (line 26). 

Multifaceted strategy

Comment 4. Line 127: since there are very many POCT on the market, I suggest to replace ‘the’ by ‘a’ before ‘point-of-care testing’. 

Reply: Thank you for your suggestion. We have removed 'we' from the sentence but did not replace it with anything (line 130). Using 'a' would create an awkward combination with the abbreviation, hence our choice to leave out both.

Results

Comment 5. Legend of Fig 2. (line 220): I suggest to add ‘and test results’ 

Reply: We share your view and have added ‘and test results’ (line 223). 

Comment 6. Line 234: Please provide some more information on ‘short turnaround time’: what was the median time and range in minutes? 

Reply: Test results were read after 20 minutes. We did not keep exact track of how quickly the doctor communicated the results, but it was immediately after. Therefore, we added, ‘of approximately 20 minutes’ (line 238).

Discussion

Comment 7. Line 300: please adjust and do not start a new sentence. So ‘was made, for example….’.

Reply: We have adopted this suggestion (line 305). 

Comment 8. Line 303: please adjust to: ‘non-participants’.

Reply: Thank you for your suggestion. We have adjusted ‘non-participators’ to ‘non-participants’ (line 309). 

Comment 9. Line 326: Please replace ‘whom’ by ‘who’.

Reply: This has been adjusted in the manuscript (line 331).

Comment 10. Line 332: Please adjust to ‘successful’.

Reply: Thank you, this has been revised (line 337). 

Limitations

Comment 11. Although the DETERMINE test for HBsAg and the OraQuick assay for HCV antibody detection are both stated to have a very high sensitivity, would you still consider using these in future settings on finger prick blood?

Reply: Thank you for your question. These tests were easy to administer and all tests were performed successfully, so yes we would consider using these in future settings on finger prick blood. However, we acknowledge that the tests do not provide information on passed HBV infections, so no statements can be made about this within this population. Although we already covered this very briefly, we have elaborated a little more on this: ‘’Nevertheless, we lack certainty due to non-participation and lack of information about their cleared HBV infections hepatitis status., as the HBV test did not provide information on this.’’ (line 310-311). 

Comment 12. Also, I think it is a pity that no tests seem to have been performed concomittantly on regular EDTA blood samples (as was described to be performed in case confirmation was needed) and in the finger prick blood samples with the POCTs, to have an impression of the actual validity of the high sensitivity of these POCTs in this setting.

Reply: We agree with this observation. Ideally, we would confirm the test results, but for budgetary reasons, we could not do a validity study alongside it. 

Reviewer 2.

In this manuscript, the authors aimed to reach Syrian migrants as part of a nationally coordinated implementation research to reach first-generation non-European migrants for HBV and HCV screening using point-of-care testing in two Dutch municipalities (Heerlen and Maastricht). The article described a solid and very well-presented research which is absolutely consonant with OMS agenda for viral hepatitis elimination by 2030.

Overall, the study had great merit in the detailed presentation of the multifaceted strategies used that made it possible to reach a significant number of migrants (approximately 1/3 of the individuals invited to participate), an extremely valuable information that could be implemented in similar studies in different parts of the world in an effort to achieve the microelimination of viral hepatitis in key populations. Although previous studies had reported HBV and HCV infection rates over 2% in Syrian migrants, no positive cases were detected within the 269 individuals screened. Nonetheless, based on a previous systematic literature review that supported the multifaceted strategies proposed by the authors, the research efforts fulfilled their objective of screening a difficult-to-reach population, addressing one of the main bottlenecks for the elimination of viral hepatitis.

The quality of the written English is excellent thorough out the article and, in my opinion, only a few minor revisions are necessary for publication:

Comment 1. Page 13, line 277: A parenthesis ")" mistakenly appeared before the beginning of the sentence.

Reply: Thank you for your observant eye. The parenthesis is deleted (line 281).

Comment 2. Page 13, line 278: An "n" was missing before “=3” in the parenthesis showing the number of people who reported lack of knowledge about hepatitis.

Reply: Thank you, this has been added (line 282). 

Comment 3. Page 13, line 291: There is an error in the caption of figure 3: part(non)participation is written instead of (non)participation.

Reply: Again, thank you very much for paying attention to these textual errors. ‘part’ has been deleted (line 295). 

Comment 4. Page 15, lines 331 and 333: I believe the in-text citations of references 34 and 35 are not correctly presented and should be corrected.

Reply: We share your view and have adapted these in-text citations (line 336 and line 339), and also applied this to line 329.

Additional Editor Comments:

Dear Author,

I have read the paper and comments of the reviewers. After careful reading, I observed that sampling strategy adopted by the authors is not so good, the optimal alternative should be RDS, Starfish, etc. However, authors should explain why they choose this sampling method, they may show that a less than optimal technical choice could be better once the relative costs are taking in consideration. Please also read and answered the reviewer's comments.

Sincerely,

Livia

Response: Dear Editor, thank you very much for reviewing our manuscript and for your observations. We value the thoughtful comments provided by you and the reviewers. To arrive at an appropriate sampling strategy for this population, focus groups were conducted. We elaborate on this in lines 104-110: ‘‘A Syrian physician employed by the PHS undertook focus groups with Syrian migrants, conducting separate sessions for men and women. The outcomes of these sessions revealed the participants’ preference for an invite by mail through an address file, sent by a trusted organisation that provided a clear rationale for the free screening. The municipality was perceived as reliable, attributed to their role in providing housing for migrants. Therefore, the migrants were invited by letter with a clear reason for the invitation from the PHS - an organization clearly related to the municipality.’’. 

As the Syrian migrants in the focus groups preferred this, we chose to commit to this sampling strategy. 

If a comment is not addressed to your satisfaction, please let us know. Once more, we would like to thank you for your time, positive feedback, and valuable insights. We look forward to the rest of the process.

Thank you in advance. 

Kind regards, 

On behalf of all authors,

Chrissy Moonen

---

## [Decision Letter · Decision Letter 1]

1 Oct 2024

PONE-D-23-37923R1Reaching Syrian migrants through Dutch municipal registries for hepatitis B and C point-of-care testingPLOS ONE

Dear Dr. Moonen,

Thank you for submitting your manuscript to PLOS ONE. After careful consideration, we feel that it has merit but does not fully meet PLOS ONE’s publication criteria as it currently stands. Therefore, we invite you to submit a revised version of the manuscript that addresses the points raised during the review process.

We look forward to receiving your revised manuscript.

Kind regards,

Livia Melo Villar

Academic Editor

PLOS ONE

Additional Editor Comments:

Dear Authors,

Thanks for sending this revision, some of the reviewers raised out some questions and I agree with him. Please could revise the paper as suggested by these reviewers,

Sincerely,

Livia

Reviewers' comments:

Reviewer's Responses to Questions

**Comments to the Author**

1. If the authors have adequately addressed your comments raised in a previous round of review and you feel that this manuscript is now acceptable for publication, you may indicate that here to bypass the “Comments to the Author” section, enter your conflict of interest statement in the “Confidential to Editor” section, and submit your "Accept" recommendation.

Reviewer #1: All comments have been addressed

Reviewer #3: (No Response)

Reviewer #4: All comments have been addressed

2. Is the manuscript technically sound, and do the data support the conclusions?

Reviewer #1: (No Response)

Reviewer #3: Partly

Reviewer #4: Yes

3. Has the statistical analysis been performed appropriately and rigorously? 

Reviewer #1: (No Response)

Reviewer #3: N/A

Reviewer #4: Yes

4. Have the authors made all data underlying the findings in their manuscript fully available?

Reviewer #1: (No Response)

Reviewer #3: Yes

Reviewer #4: No

5. Is the manuscript presented in an intelligible fashion and written in standard English?

Reviewer #1: (No Response)

Reviewer #3: Yes

Reviewer #4: Yes

6. Review Comments to the Author

Reviewer #1: (No Response)

Reviewer #3: Authors: I was not among the reviewers of the original manuscript but was just asked to review the R! version. Comments made here and in the attached copy of the manuscript reflect a de novo reading and are not made in light of any comments made by previous reviewers (which I have read). The work is well conceived and has useful contents. The principle objective "reaching Syrian migrants...for HBV and HCV...testing" is formulated in the context of HBV/HCV significance in the public health context. The approach taken and its results, 32% participation (with no infections identified) are the core of the project. Examining the solicitation ("reaching") as described, leaves a number of questions about just what was done (specific language, description, and questions) that are identified as comments in the manuscript copy attached. More complete details would enable analyzing the effectiveness of the approach and could lead to more specific recommendations for how response might be improved. However, recognizing that no infections were identified in ca. 250 of 850 individuals solicited against an expected ca. 2% +/-, conclusions regarding efficacy of this whole population approach are appropriate. The value of the manuscript to others will be increased by providing more detail on the specific methods used in solicitation...and analysis that may support speculation on what alternatives should be considered.

Reviewer #4: The second version of this paper addresses the comments received during peer review, and the changes adopted provide further clarification

It shows improvement and the authors are to be commended.

7. PLOS authors have the option to publish the peer review history of their article (what does this mean?). If published, this will include your full peer review and any attached files.

Reviewer #1: **Yes: **Sylvia Maria Bruisten

Reviewer #3: No

Reviewer #4: No

---

## [Author Response · Author response to Decision Letter 1]

10 Oct 2024

Dear reviewer,

Thank you for the additional reviewer comments. Below you can find our response per reviewer comment. Language suggestions have been adopted and are addressed in the PDF file of reviewers' comments. For the sake of clarity, these suggestions have not been reproduced in this document.

Reviewer comment 1: Can this group be described more fully? Relevant to the first study objective, is the group identifiable in some national or local database?

Authors response: We added 'identified through municipal registries' at the end of the sentence.

Reviewer comment 2: Were the focus group participants from the two communities?

Authors response: No, Syrian migrants from other municipalities also participated in the focus groups. 

Reviewer comment 3: What reason? This is a critical feature related to the 1st study objective. Perhaps include the wording used in the letter in an appendix or figure.

Authors response: After this sentence, we added: 'The invitation letter explained that asymptomatic infections may go undetected and that testing could clarify the infection status and enable timely treatment. It also stressed the importance of protecting others from the infectious nature of HBV and HCV and noted the higher prevalence of these viruses in the country of birth as the reason for the invitation.'

Reviewer comment 4: How was the preference indicated? Was a response requested in the invitation letter...or was the preference determined from the focus groups?

Authors response: We changed 'respondents' to 'participants of the focus groups'

Reviewer comment 5: awareness of what, specifically? Seriousness of HBV/HCV infection?

Authors response: We changed 'awareness' to ‘knowledge about the viruses’.

Reviewer comment 6: The term screening is use here referring to the actual administering of the HBV/HCV tests. The term is not used consistently., using screening and testing interchangably. Confusion would be avoided by consistency.

Authors response: Thank you for the critical eye. We agree and therefore changed 'screening' to 'testing' throughout the whole document. 

Reviewer comment 7: Information on what, specifically? Information on this study? ...Information on the importance and availability of HBV/HCV screening

Authors response: Added: 'on the viruses and the testing opportunity'

Reviewer comment 8: Was this information provided in the original contact letter? e.g. Testing will sill consist of a finger pric performed by the PHS trained and certified nurses at the (specific local?) PHS facility?

Authors response: No, this information was not provided in the original invitation letter. 

Reviewer comment 9: Was assurance of data confidentiality provided to prospective participants in initial contact?

Authors response: It was not mentioned in the invitation letter, but it was mentioned at the registration desk. Therefore, we added a sentence: 'During registration, PHS staff explained the use of a participant code to the participants'.

Reviewer comment 10: Was assurance of data confidentiality provided to prospective participants in initial contact?

Authors response: It was not mentioned in the invitation letter, but it was mentioned at the registration desk. Therefore, we added a sentence: 'During registration, this was explained to the participant by PHS staff'. 

Reviewer comment 11: As suggested in previous comments, the details of this invitation...resulting from the focus group process?...is a key element that must have had a significant (not suggesting a statistical test) bearing on the level (32%) of participant response. More detail on the invitation is important.

Authors response: More information on the invitations is added in line (line 110-114 in the marked-up version): ‘'The invitation letter explained that asymptomatic infections may go undetected and that testing could clarify the infection status and enable timely treatment. It also stressed the importance of protecting others from the infectious nature of HBV and HCV and noted the higher prevalence of these viruses in the country of birth as the reason for the invitation.'

Reviewer comment 12: This is not insignificant information asked of a participant. Were potential participants informed that they would be asked to provide such information in the original invitation...or in subsequent contact? Again, this seems to be important to interpreting the success of the first study objective.

Authors response: At the PHS it is standard practice to complete a questionnaire on medical history before a medical procedure, so this was not communicated in advance. 

Reviewer comment 13: Presumably, this information was also available on all 832 (see lines 133-134 of text). A comparison of the demographic characteristics of the non respondents to the respondents would be useful in understanding the reachability of this kind of target population.

Authors response: Information on sex, age and municipality of all potential participants was available through the municipal registers. The characteristics of both participants as non-participants have been reported in table 2. 

Reviewer comment 14: could have been HAV...any information on whether the respondents meang HBV or HCV?

Authors response: Yes, but most did not know what type as reported. 

Reviewer comment 15: 23 out of 832-269=563 is about 4%. Doesn't provide much information on why the effort to reach the population was not more successful.

Authors response: We think this provides some information on non-participation and as this often is not taken into account in other studies, we feel that it is worth mentioning. 

Reviewer comment 16: Characterizing 32% participation as "relatively high" should not be accepted without justification. In a population expected to have ca. 2% HBV/HCV positive, soliciting participation from 850 might produce 16 positives (850 x 0.02 = 16). If only 32% participate one might expect 5 positives. Not much room for variation. If the objective is "Reaching Syrian migrants..." and 68 % of those contacted didn't participate, there's a lot of room for improving the approach.

Authors response: T Thank you. We have removed this sentence and have added: 'For this method, we achieved a relatively high participation rate (32.3%), compared to studies with similar recruitment methods.' After this, we copy-pasted the second paragraph. The next paragraph starts with 'The usefulness of this strategy is still considered limited'. 

Reviewer comment 17: Perhaps not...68% were not reached.

Authors response: When looking at studies with comparable recruitment methods, we have achieved a relatively high participation rate. Therefore, we have rephrased this into: 'For this method, we have achieved a relatively high participation rate (32%), compared to studies with similar recruitment. For example.....'

Reviewer comment 18: The data presented indicates that the study method...letter invitations...resulted in 63% of participants over 30, with a similar proportion (58%) of non-particiapants over 30.

Authors response: Thank you for your suggestion. We do not feel that this adds to the essence and clarity of this paragraph and therefore we did not adopt this suggestion. 

Reviewer comment 19: Is it reasonable to say that it was preferred if venepuncture was not offered as an alternative?

Authors response: Thank you. We removed this because it was indeed not offered and the preference was not monitored. 

Reviewer comment 20: ...Hamed (35). Migrants... (need a space)

Authors response: This was already adapted in the previous revision. 

Reviewer comment 21: Does this mean something more like... Migrants may not access available health care to the same degree as the native population due to lack of knowledge, language, and cultural factors (35).

Authors response: Thank you for your suggestion. We have rephrased this as follows: 'Compared to natives, migrants may face more challenges in accessing healthcare, for example, due to linguistic barriers or discrimination.'

Reviewer comment 22: The interest in identifying HBV/HCV infections in a population of migrants from a region of significantly higher prevalence than in the native population is justified. In this study, soliciting participation in a free testing program produced only 32% of about 850 solicited resulting in identifying no HBV or HCV infections. This suggests a more efficient means is needed that testing is needed. Focussing testing on the higher risk segments of the population should be more efficient (i.e. targeted testing...IVD users, etc)

Authors response: Thank you for your suggestion. We agree that additional efforts are needed to reach more people at risk. Therefore, we have rephrased this: 'With about one-third participation rate, we can conclude that a multifaceted strategy for HBV and HCV testing, including an invite by letter, is a good way to reach migrants for testing, but additional efforts are needed to reach more people at risk.’ Furthermore, we deleted the recommendations in the conclusion, as they were already mentioned in the recommendations. 

The revision of the manuscript have been submitted. Thank you for your time and efforts. 

Kind regards,

On behalf of all the authors,

Chrissy Moonen

---

## [Decision Letter · Decision Letter 2]

16 Dec 2024

Reaching Syrian migrants through Dutch municipal registries for hepatitis B and C point-of-care testing

PONE-D-23-37923R2

Dear Dr. Moonen,

We’re pleased to inform you that your manuscript has been judged scientifically suitable for publication and will be formally accepted for publication once it meets all outstanding technical requirements.

Kind regards,

Livia Melo Villar

Academic Editor

PLOS ONE

Additional Editor Comments (optional):

Reviewers' comments:

Reviewer's Responses to Questions

**Comments to the Author**

1. If the authors have adequately addressed your comments raised in a previous round of review and you feel that this manuscript is now acceptable for publication, you may indicate that here to bypass the “Comments to the Author” section, enter your conflict of interest statement in the “Confidential to Editor” section, and submit your "Accept" recommendation.

Reviewer #1: All comments have been addressed

Reviewer #3: All comments have been addressed

2. Is the manuscript technically sound, and do the data support the conclusions?

Reviewer #1: Yes

Reviewer #3: Partly

3. Has the statistical analysis been performed appropriately and rigorously? 

Reviewer #1: Yes

Reviewer #3: N/A

4. Have the authors made all data underlying the findings in their manuscript fully available?

Reviewer #1: Yes

Reviewer #3: Yes

5. Is the manuscript presented in an intelligible fashion and written in standard English?

Reviewer #1: Yes

Reviewer #3: Yes

6. Review Comments to the Author

Reviewer #1: I have no further comments. I was already pleased with the R1 rebuttal version. Only the other reviewer still had questions to be answered.

Reviewer #3: This reviewer appreciates the authors response with description of reasoning to comments and suggestions made to the R1 manuscript version, resulting in revised R2 manuscript. It is essential in conveying to an interested reader the information assembled in the text that it be described objectively. The value of a negative finding supported by the data is as great as would be a positive finding.

7. PLOS authors have the option to publish the peer review history of their article (what does this mean?). If published, this will include your full peer review and any attached files.

Reviewer #1: **Yes: **Sylvia M. Bruisten

Reviewer #3: **Yes: **Jerry E. Ongerth

---

## [Editor Report · Acceptance letter]

7 Jan 2025

PONE-D-23-37923R2 

PLOS ONE

Dear Dr. Moonen, 

I'm pleased to inform you that your manuscript has been deemed suitable for publication in PLOS ONE. Congratulations! Your manuscript is now being handed over to our production team.

Kind regards, 

on behalf of

Dr. Livia Melo Villar 

Academic Editor

PLOS ONE